environmental science/environmental engineering

anammox, denitrification, salinity, granular sludge, microbial community

**Author for correspondence:**
Zhaozhao Wang
e-mail: w-z-z@163.com

This article has been edited by the Royal Society of Chemistry, including the commissioning, peer review process and editorial aspects up to the point of acceptance.

# Effects of salinity on the simultaneous anammox and denitrification process: performance, sludge morphology and shifts in microbial communities

Zhaozhao Wang[1,2], Peng Gao[1,2], Ying Ji[1,2], Huan Zhang[1,2], Xinjuan Wu[1,2], Jun Ma[1,2] and Simin Li[1,2]

[1]College of Energy and Environmental Engineering, and [2]Hebei Technology Innovation Center for Water Pollution Control and Water Ecological Remediation, Hebei University of Engineering, Handan 056038, People's Republic of China

ZW, 0000-0002-0507-7205

In this study, the long-term effects of different salinities on the performance, sludge morphology and shifts in microbial communities were studied in a simultaneous anammox and denitrification (SAD) process at a C/N ratio of 0.5. Stable nitrogen removal efficiencies of 86.96 and 84.58% and nitrogen removal rates of 0.95 and 0.93 kg $(m^3 d)^{-1}$ could be achieved under low (25 mmol $l^{-1}$) and moderate (50 mmol $l^{-1}$) salinity, respectively. However, the performance collapsed when the system was exposed to high salinity (100 mmol $l^{-1}$). The content of extracellular polymeric substances increased as salinity increased, which resulted in larger sizes of granular sludge under low and moderate salinities. Nevertheless, high salinity shock disintegrated granular sludge, thereby decreasing the average granule size. The Illumina-Miseq sequencing results revealed that *Candidatus Jettenia* was the sole salinity-tolerant AnAOB genus during the entire operation, whereas the main denitrification bacterial genera shifted from *Denitrisoma* under low salinity to *Denitrisoma*, *Thauera* and *Ignavibacterium* under high salinity. The results of this study provide a comprehensive and practical evaluation of the SAD process for organic nitrogen-rich saline wastewater treatment.

# 1. Introduction

The eutrophication of slow-flowing water bodies resulting from excessive nitrogen discharge has attracted much attention [1,2]. Consequently, finding ways to remove nitrogen economically and efficiently to improve the water environment has become a major focus of research. The conventional biological nitrogen removal process primarily relies on aerobic ammonia-oxidizing bacteria (AerAOB) and nitrite-oxidizing bacteria (NOB) to complete the nitrification bioreaction under oxic conditions and denitrifying bacteria (DNB) to complete the denitrification bioreaction under anoxic conditions and convert nitrogen in water bodies into dinitrogen [3]. A large amount of energy is also consumed to achieve the anoxic–aerobic alternations in the nitrogen transformation process. To optimize nitrogen removal efficacy, the addition of carbon sources as electron donors for the denitrification process is essential yet invariably results in greater quantities waste sludge, thereby increasing sewage treatment costs [4].

Recently, the anaerobic ammonia oxidation (anammox) process has become increasingly favoured for biological nitrogen removal in the wastewater treatment field. Anammox is an autotrophic nitrogen removal process occurring under anaerobic conditions that is dependent on a class of anaerobic ammonia-oxidizing bacteria (AnAOB) belonging to the phylum *Planctomycetes*, which is known to include five functional genera: *Candidatus Brocadia*, *Ca. Kuenenia*, *Ca. Scalindua*, *Ca. Anammoxoglobus* and *Ca. Jettenia* [5,6]. Nevertheless, AnAOB have a slow growth rate and can be affected by both substrates and environmental factors, among which organic matter is extremely important. As typical autotrophic bacteria, AnAOB activity is prone to being inhibited by various organic environments [7,8]. Most practical wastewater usually contains both organic and nitrogenous sources, which have limited applications in the anammox process for organic nitrogen-rich wastewater treatment. In addition, nitrate, which accounts for approximately 10% of total nitrogen (TN), is produced after the anammox bioreaction [9]. To alleviate the effects of the inhibition of organic matter on the anammox process, researchers have introduced DNB to remove organic matter and nitrate via the denitrification process to achieve a simultaneous anammox and denitrification (SAD) process. Nitrite has also been found to accumulate during the denitrification process under low C/N ratios and could thus provide electron acceptors for the anammox pathway. The principal bioreactions involved in the SAD process are depicted in equations (1.1)–(1.3):

Anammox:

$$NH_4^+ + 1.32NO_2^- + 0.066HCO_3^- + 0.13H^+ \rightarrow 1.02N_2 + 0.26NO_3^- + 0.066CH_2O_{2.5}N_{2.5} + 2.03H_2O. \tag{1.1}$$

Denitrification:

$$6NO_3^- + 2CH_3OH \rightarrow 6NO_2^- + 2CO_2 + 4H_2O \tag{1.2}$$

and

$$6NO_2^- + 3CH_3OH \rightarrow 3N_2 + 3H_2O + 6OH^- + 3CO_2. \tag{1.3}$$

He *et al.* [10] successfully optimized the coupling of anammox and denitrification processes, which could achieve a nitrogen removal efficiency of 90% and a chemical oxygen demand (COD) removal efficiency of 90%. Qin *et al.* [11] introduced glucose to the anammox process to initiate the SAD process and found that $NO_2^- - N$ was provided to the anammox process via a partial-denitrification process. The SAD process thus shows great potential to be applied to organic nitrogen-rich wastewater treatment.

Generally, the composition of practical domestic or industrial wastewater is complex. Some types of wastewater contain salt (e.g. pickle industrial wastewater [12], seafood industrial wastewater [13], of which the salinity concentration is normally in the range of 170–2500 mmol $l^{-1}$), which creates another challenge during the SAD process. High salt content places stress on the microbial flora, inhibits the activity of key enzymes and eventually leads to cell disintegration and death [14]. In addition, appropriate acclimation strategies (e.g. dilution of the original saline wastewater, salinity-step domestication, etc.) can be used to permit the biological process to effectively treat saline wastewater under the salinity threshold.

The effects of salinity on anammox or the denitrification process (two key nitrogen removal pathways in the SAD process) have been reported. Some researchers have employed the anammox process for nitrogen-rich saline wastewater treatment and found that the stability of the process was largely dependent on the inoculated AnAOB types and salinity concentrations [15]. In general, the adaptation

period is much longer for freshwater anammox bacteria (FAB) than for marine anammox bacteria (MAB). The salinity tolerance threshold of the FAB can reach the same level as MAB after sufficient acclimation. Second, salinity has a direct impact on AnAOB activity and sludge properties. The inhibition of AnAOB activity stemming from high salinity was mainly driven by the weakening of intracellular carbohydrate and lipid metabolism, the blockage of the intracellular energy supply and the reduction of membrane transport capacity [16]. The sludge morphological features, surface functional groups and metabolic production are also affected by salinity through microbial responses [17].

Furthermore, salinity alters the structure of microbial communities and the dominant flora participating in the anammox process. For example, Jeong et al. [18] found that salinity could induce the overgrowth of heterotrophic bacteria during the autotrophic nitrogen removal process, which could cause the failure of anammox systems. Wu et al. [19] found that AnAOB diversity might significantly change because of the discrepancy in their salinity-driven growth kinetics. Lu et al. [20] found that increased salinity (greater than 160 mmol l$^{-1}$) resulted in a shift from Candidatus kuenenia to unclassified Brocadiaceae as the dominant AnAOB genus.

Similarly, salinity has a substantial impact on the performance, nitrogen removal pathway and microbial community structure in the denitrification process. Zhai et al. [21] found that the denitrification efficacy decreased as a result of the inhibition of microorganismal activities by high salinity (598.9 mmol l$^{-1}$); moreover, the uncompleted denitrification stemming from the nirS and nirK genes, which are responsible for the further transformation of the nitrite, significantly decreased under high salinity. Miao et al. [22] reported that salinity significantly affected the microbial community in the denitrification process; Marinobacter and Halomonas were the prevailing DNB genera for nitrate removal under high and low salinity conditions, respectively. Nevertheless, Thauera was identified as a functional genus of bacteria in a partial-denitrification granule sludge system for nitrite production under high salinity conditions [23].

To date, few studies have examined the effects of salinity on the SAD process. Li et al. [24] achieved simultaneous C and N removal using a 'Candidatus Brocadia sinica'-dominated SAD process through seawater step-up domestication. Furthermore, this same research group has developed an enhanced strategy (mannitol addition) for the SAD process and established substrate inhibition models under saline conditions [25]. Therefore, SAD provides a promising process for C and N removal in saline wastewater treatment. However, there is little systematic information that has been collected on the process performance, sludge properties and microbial dynamics of the SAD process in response to different salinities. Nevertheless, there is a need to establish the adaptation and stability laws of this process under different degrees of salinity stress to assess the feasibility of using the SAD process in saline wastewater treatment.

In this study, the long-term effects of different degrees of salinity on the SAD process were explored. Specifically, the aims were to (i) evaluate the nitrogen and carbon removal performance; (ii) characterize the physio-chemical and biochemical properties of the granular sludge; (iii) analyse the dynamic changes in microbial community structure and functional bacteria in the system; and (iv) assess the salinity tolerance of the SAD process to provide theoretical and technical support for the application of the SAD process in organic nitrogen-rich saline wastewater treatment.

# 2. Material and methods

## 2.1. Experimental materials

### 2.1.1. Experimental set-up

The continuous-flow experimental device was a 10 l (working volume) upflow anaerobic sludge blanket (UASB) reactor composed of plexiglass with an internal diameter of 11 mm and a height of 110 mm (electronic supplementary material, figure S1a). A three-phase separator was installed on the top of the reactor to separate the sludge, effluent and gas. The reactor was equipped with a water bath cycle with two heating rods submerged in the circulation tank to maintain a constant temperature at 30 ± 1°C. The outside of the reactor was wrapped with rubber sponge insulation board to preserve heat and block light. In addition, an internal reflux was installed to control the upflow velocity. The influent sewage was pumped into the bottom of the reactor at a hydraulic retention time of 6.67 h, and the effluent flowed from the water outlet after passing through the three-phase separator.

**Table 1.** Feeding characteristics during different phases.

| phase | day | pH | $NH_4^+-N$ (mg l$^{-1}$) | $NO_2^--N$ (mg l$^{-1}$) | TN (mg l$^{-1}$) | NLR kg (m$^3$ d)$^{-1}$ | COD (mg l$^{-1}$) | NaCl (mmol l$^{-1}$) |
|---|---|---|---|---|---|---|---|---|
| I | 1–20 | 7.3–7.6 | 108.16 | 151.97 | 274.42 | 1.10 | 84.04 | 0 |
| II | 21–67 | | 106.74 | 153.48 | 274.88 | 1.10 | 84.54 | 25 |
| III | 68–134 | | 105.98 | 152.58 | 272.48 | 1.09 | 82.40 | 50 |
| IV | 135–154 | | 104.62 | 146.60 | 266.48 | 1.07 | 81.22 | 100 |
| V | 155–185 | | 97.78 | 144.63 | 254.81 | 1.02 | 78.74 | 0 |

### 2.1.2. Synthetic wastewater and seed sludge

Synthetic wastewater (dissolved organic (DO): 0–1.0 mg l$^{-1}$) was used as the feeding substrate; the ammonia, nitrite, organic matter and salinity originated from $NH_4Cl$, $NaNO_2$, sodium acetate and NaCl, respectively. Different amounts of NaCl were added into the substrates to create different salinity stresses. The concentrations of $NH_4Cl$, $NaNO_2$, sodium acetate and NaCl are listed in table 1; the other components and the trace elements are listed in electronic supplementary material, table S1. The seed granular sludge was obtained from a pilot-scale SAD-UASB reactor. The SAD-UASB reactor was successfully shifted from an anammox-UASB reactor by the following procedures: at 30°C, the anammox granular sludge were inoculated in the UASB reactor; after that, carbon source (sodium acetate) was introduced in the UASB system to cultivate the SAD granular sludge; finally, the SAD process was successfully started up by controlling the COD/NO$_x$–N ratio of 0.5 with the formations of three kinds of granules (anammox, DNB and SAD) in the bottom of the UASB reactor. The SAD-UASB was used for the treatment of high-nitrogen organic wastewater ($NH_4^+$–N: 108.2 mg l$^{-1}$; $NO_2^-$–N: 152.0 mg l$^{-1}$; $NO_3^-$–N: 14.3 mg l$^{-1}$; COD: 84.1 mg l$^{-1}$) with an average $N_2$ production of 20 l d$^{-1}$ and had been operated stably for half a year.

## 2.2. Analytical methods

### 2.2.1. Wastewater and sludge analysis

COD, $NH_4^+$–N, $NO_3^-$–N, $NO_2^-$–N, TN, mixed liquor suspended solids (MLSS) and mixed liquor volatile suspended solids were measured according to standard methods [26], and TN concentration was the sum of the $NH_4^+$–N, $NO_2^-$–N and $NO_3^-$–N concentrations. The C/N ratio was defined as the COD/NO$_x$–N ratio in the influent. The $NH_4^+$–N removal efficiency (ARE), nitrogen removal efficiency (NRE), $NO_2^-$–N removal efficiency, COD removal efficiency (CRE), nitrogen loading rate (NLR), nitrogen removal rate (NRR), COD removal rate (CRR), $\Delta NO_2^-$–N/$\Delta NH_4^+$–N ratio, $\Delta NO_3^-$–N/$\Delta NH_4^+$–N ratio, the contribution of nitrogen removal via the anammox process ($E_{anammox}$) and the contribution of nitrogen removal via the denitrification process ($E_{denitrification}$) were calculated according to the formulae in electronic supplementary material, table S2. The pH, DO and temperatures were determined by an online multiparameter sensor (Multi 3420i, WTW, Germany). The particle size of the granular sludge was determined using a standard wet screening method (aperture diameters of 3.0, 2.5, 2.0, 1.5, 1.0 and 0.5 mm) [27]. The granular sludge was placed in a culture dish (90 mm) to observe the apparent sludge morphology.

### 2.2.2. Analysis of the activities of functional bacteria

The schematic diagram of the sequencing batch reactor is shown in electronic supplementary material, figure S1b. A 250 ml serum bottle was used for the sequencing batch tests. First, the sludge mixture was collected from the UASB reactor and washed using deionized water to remove the residue. Second, the washed sludge and 250 ml of a prepared matrix were added to a serum bottle that was placed on a magnetic stirrer. After starting the magnetic stirrer, 5.0 ml of the mixture was extracted with a 10 ml syringe for centrifugation every hour, and the concentrations of $NH_4^+$–N, $NO_2^-$–N, $NO_3^-$–N, COD and MLSS were determined [28,29]. The substrates used to determine the activities of the functional bacteria are shown in electronic supplementary material, table S3. The specific AnAOB

activity (SAA) and specific DNB activity (SDA) were characterized by evaluating the specific degradation rates of TN and $NO_3^- - N$ (mgN (mgVSS d)$^{-1}$), respectively. The specific degradation rate of COD and the specific accumulation rate of $NO_2^- - N$ were also measured.

The degree to which salinity inhibited the activities of AnAOB and DNB was expressed as SAA% and SDA%, respectively. The recovery of the activities of AnAOB and DNB was expressed as $SAA_R$% and $SDA_R$%, respectively. The specific formulae were as follows (equations (2.1)–(2.4))

$$SAA\% = \frac{SAA}{SAA_0} \times 100\%, \tag{2.1}$$

$$SDA\% = \frac{SDA}{SDA_0} \times 100\%, \tag{2.2}$$

$$SAA_R\% = \frac{SAA_R}{SAA_0} \times 100\% \tag{2.3}$$

and

$$SDA_R\% = \frac{SDA_R}{SDA_0} \times 100\%, \tag{2.4}$$

where $SAA_0$ and $SDA_0$ represent the initial maximum activities of AnAOB and DNB when no salt is added; SAA and SDA represent the activities of AnAOB and DNB at a specific salinity, respectively; and $SAA_R$ and $SDA_R$ represent the recovered activities of AnAOB and DNB after the elimination of salinity stress.

### 2.2.3. Extracellular polymeric substances extraction and Fourier-infrared spectroscopy analysis

At the end of each phase, anammox granular sludge was taken from the reactor to determine the EPS content, and the loose-bound extracellular polymeric substances (LB-EPS) and the tight-bound EPS (TB-EPS) were extracted following a modified heating method [30]. The LB-EPS and TB-EPS were characterized as the sum of the carbohydrate and protein fractions. The PN and PS fractions of the LB-EPS and TB-EPS were determined using the modified Bradford method with bovine serum albumin as the standard and the anthrone-sulfuric acid method with glucose as the standard, respectively [31].

The LB-EPS and TB-EPS extractions were frozen in the cold trap of a freeze-drying machine (ESW-DG-T, Shanghai Antler Research Instruments Co. Ltd, China) for 4–6 h and then dried. The freeze-dried sample and potassium bromide (KBr) were pressed at a ratio of 1 : 100, and spectral analysis was conducted using a Fourier-infrared spectrometer (FT-IR Affinity-1S, Shimadzu Company, Japan). The collected data were processed and analysed using Origin 8.5 software.

### 2.2.4. High-throughput sequencing

Sludge samples (10 ml for each sample) were collected at the end of each phase, and a total of five samples were collected (S1, S2, S3, S4 and S5). Microbial community genomic DNA was extracted from the five samples using the E.Z.N.A.® soil DNA Kit (Omega Bio-tek, Norcross, GA, USA). The hypervariable region V4 of the bacterial 16S rRNA gene was amplified with primer pairs 515F (5′-GTGCCAGCMGCCGCGG-3′) and 806R (5′-GGACTACHVGGGTWTCTAAT-3′) [32]. An Illumina-MiSeq PE300 platform (Shanghai Majorbio Bio-Pharm Technology Co. Ltd) was used for sequencing, and the raw reads were deposited into the National Center for Biotechnology Information Sequence Read Archive database (Accession number: SRP 239512). The operational taxonomic units (OTUs) with a 97% similarity cut-off were clustered using UPARSE (v. 7.1, http://drive5.com/uparse/). The mapping analysis was performed with Origin 8.5 software to determine the microbial diversity.

# 3. Results and discussion

## 3.1. The effect of different salinities on the SAD process

### 3.1.1. Process performance and functional bacteria activities

The average concentrations of $NH_4^+ - N$, $NO_2^- - N$, $NO_3^- - N$ and COD in the influent were maintained at 104.89, 150.84, 14.04 and 82.38 mg l$^{-1}$ throughout the entire operation of the reactor, respectively. The responses of the performance and functional bacteria activities in the SAD process under different salinities are shown in figure 1 and electronic supplementary material, figure S2, respectively.

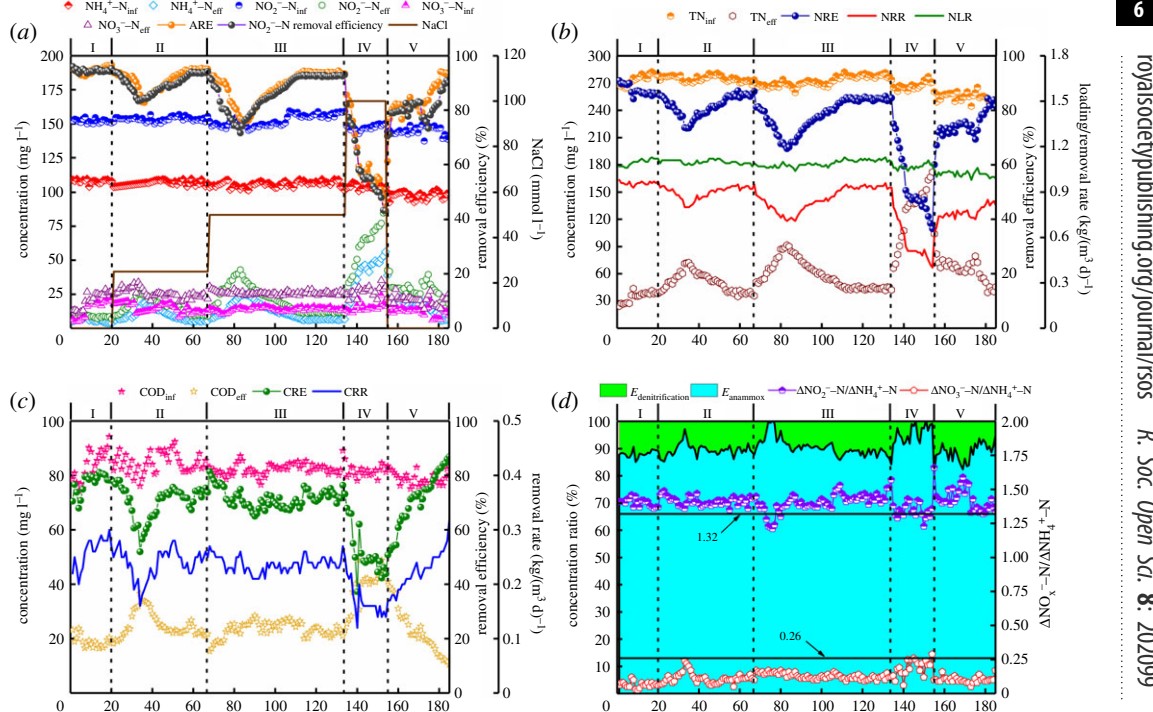

**Figure 1.** Performance of the SAD process during different operational phases. (*a*) Ammonia, nitrite and nitrate removals; (*b*) TN removal; (*c*) COD removal and (*d*) changes in $\Delta NO_2^--N/\Delta NH_4^+-N$ and $\Delta NO_3^--N/\Delta NH_4^+-N$.

In phase I (no salinity addition), the SAD process exhibited stable nitrogen and COD removals. Specifically, 87.35% of TN and 77.28% of COD could be removed under an NLR and CLR of 1.10 and 0.34 kg $(m^3 d)^{-1}$, respectively. Moreover, the coupling process was dominated by the anammox pathway ($E_{anammox}$ of 88.11%) with an SAA of 0.31 mg $(mgVSS d)^{-1}$. The denitrification pathway ($E_{denitrification}$ of 11.89%) was an effective supplement for nitrogen removal with an SDA of 0.39 mg $(mgVSS d)^{-1}$ VSS d. The stable stoichiometry ratio of $\Delta NO_2^--N/\Delta NH_4^+-N$ and $\Delta NO_3^--N/\Delta NH_4^+-N$ was maintained at 1.40 and 0.06, respectively, which further indicated that the SAD process ran smoothly. This observation was consistent with a previous study [33] showing that the $\Delta NO_2^--N/\Delta NH_4^+-N$ ratio was higher and the $\Delta NO_3^--N/\Delta NH_4^+-N$ ratio lower than the corresponding theoretical value (1.32, 0.26) of the anammox process.

In phase II (salinity increased to 25 mmol $l^{-1}$), a typical 'concave curve'-style performance was observed for nitrogen and COD removal. In the initial 15 days, ARE, NRE and CRE decreased to 82.94, 73.49 and 56.59% and increased gradually to 95.1, 86.96 and 73.82% in the following 32 days. These results suggested that low salinity had a recoverable impact on the SAD process for nitrogen and COD removal. Moreover, the SDA% decreased to 66.67%, which was a bit lower than SAA% (74.19%). This finding further revealed that DNB were more sensitive to salinity than AnAOB, which was reflected by the decreased $E_{denitrification}$ (by 10.11%) and the increased $E_{anammox}$ (89.49%).

In phase III (salinity increased to 50 mmol $l^{-1}$), a similar nitrogen removal trend was observed. ARE and NRE decreased to 76.9% and 65.92%, respectively, and gradually recovered to 93.44% and 84.58%, respectively. ARE and NRE took longer (approx. 51 days) to return to their initial levels in this phase. The recovery rate of ARE was higher in this phase than in phase II, which reflected the increased SAA (0.26 mgN $(mgVSS d)^{-1}$) after salinity shock and acclimation. This result suggested that AnAOB exhibited strong salinity tolerance despite the stronger salinity shock. CRE slightly decreased in the first 10 days and then stabilized at approximately 70.78%. This strange phenomenon possibly stemmed from the induced growth of the other halophilic heterotrophic bacteria through salinity introduction, which increased the consumption of organic matter. The reduction in available organic matter as electron donors further strengthened the inhibition of DNB except for the negative effects from increased salinity. In addition, the SDA was reduced by 0.18 mgN $(mgVSS d)^{-1}$ and $E_{denitrification}$ by 9.50%.

In phase IV (salinity further increased to 100 mmol $l^{-1}$), process performance collapsed, as NRE and CRE decreased to 36.51% and 43.86%, respectively, during 20 days of operation. The increased inhibition

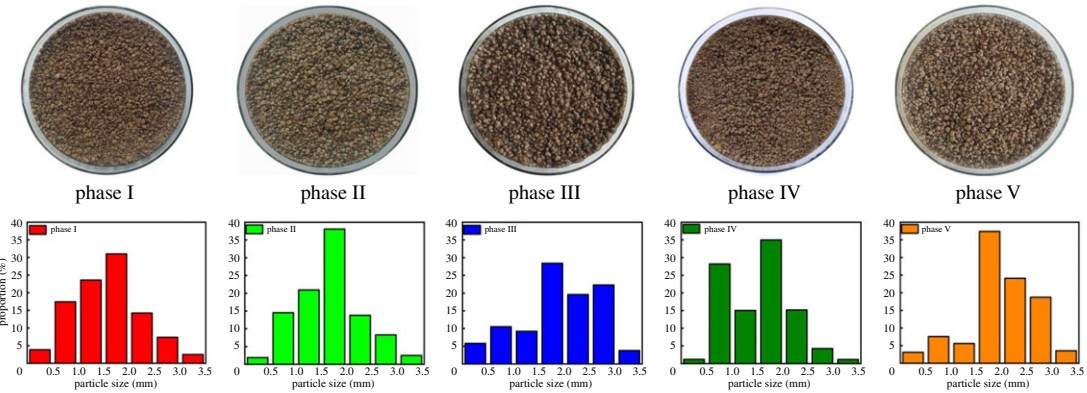

**Figure 2.** Morphological features and particle size distributions of granular sludge during different operational phases.

of AnAOB (0.13 mgN (mgVSS d)$^{-1}$) and DNB (0.13 mgN (mgVSS d)$^{-1}$) could explain this unexpected result. Part of the functional bacteria might be severely dehydrated and result in a loss in enzyme activity under high salinity conditions. The lower SDA% (33.33%) and the increased specific nitrite accumulation rate (0.08 mgN (mgVSS d)$^{-1}$) suggested that there was a shift in the flora of DNB that could not achieve complete denitrification. The decreased $\Delta NO_2^- - N / \Delta NH_4^+ - N$ (1.34) and increased $\Delta NO_3^- - N / \Delta NH_4^+ - N$ (0.19) approached theoretical values of the anammox bioreaction, suggesting that the denitrification pathway had been further weakened. This result was consistent with the decreased $E_{denitrification}$ (4.27%).

To recover the functionality of the SAD system, salinity was no longer added into the substrate in phase V. After 30 days of operation, the NRE (93.43%) and CRE (90.08%) were obtained. Furthermore, the SAA and SDA recovered to 0.27 and 0.18 mgN (mgVSS d)$^{-1}$, respectively, with SAA$_R$% and SDA$_R$% values of 87.10% and 46.15%, respectively. These results suggested that the activities of AnAOB and DNB could self-recover after high salinity shock.

Overall, the SAD process could resist the low and moderate salinity shock, although a lack of stability in performance was also observed. High salinity had an irreparable impact on the SAD system stemming from the dramatically decreased AnAOB and DNB activities. This deteriorating effect could be relieved after the salinity stress was eliminated.

## 3.2. Sludge characteristics

### 3.2.1. Sludge morphology

Sludge morphology is affected by salinity to various degrees because of the toxic responses of microbes [34]. The apparent morphology and particle size distribution of granular sludge during different phases are shown in figure 2. In phase I, the granular sludge was dark red. Generally, three types of granular sludge coexisted in the SAD system, including AnAOB granules, DNB granules and SAD granules embedded by AnAOB inside with DNB attached outside [35]. Particle sizes in the range of 1.5–2.0 mm accounted for the highest proportion of particles (31.02%). Particle sizes greater than 3.0 mm and below 0.5 mm accounted for 2.50% and 3.85% of all particles, respectively. Overall, the particle size of granular sludge approximated a normal distribution.

In phase II, the apparent colour of granular sludge was dark brown, and the average particle size increased as the salinity increased. The proportion of granular sludge with particle sizes below 1.0 mm decreased to 16.41%. A similar pattern was also observed in phase III. More regular, spherical granular sludge tended to have smoother surfaces and darker colours. The proportion of particle sizes over 1.5 mm increased to 62.65%, which made the average particle size reach its maximum value. These results probably stemmed from the increased EPS production that accelerated sludge granulation under tolerated salinity [36]. Larger sizes of granular sludge had stronger resistance to salinity toxicity by offering interior functional bacteria a solid shelter [37].

Nevertheless, the average particle size of granular sludge significantly decreased as the proportion of particles below 1.5 mm increased, accounting for 44.42% of the particles after the sudden strong salinity shock in phase IV. The cause of the disintegration of a few granular sludge particles was from the death of bacterial cells under high osmotic pressure [34]. Because of its external location, which is more

vulnerable to experiencing the toxic effects of salinity, DNB detached from the SAD granular sludge, thereby converting the larger granules into smaller ones. Thus, the inner AnAOB of granular sludge were exposed to the outside, which made the granules appear red. These results were consistent with the corresponding collapse in process performance. The reconstruction of broken granules was assumed to not be possible because of the continued loss of bacteria activities if high salinity stress was present for long periods. In the recovery period (phase V), the apparent colours of the granular sludge were dark red, which resulted from the recovery of the functional bacteria activities. Moreover, the proportion of granular sludge with particle sizes over 1.5 mm accounted for 83.74% of all particles, which revealed that the disintegration of the granular sludge resulting from the strong salinity shock was recoverable. It should be noted that no sludge was discharged out of the reactor throughout the experiment, which resulted in a relatively stable sludge concentration, and thus supported the recovery performance of the system.

In sum, the sludge morphology was affected by salinity to various degrees. The characteristics of granular sludge were strongly dependent on salinity concentrations. The re-aggregation behaviour of disintegrated granules after the elimination of high salinity stress possibly stemmed from the quorum sensing of the surviving functional bacteria.

### 3.2.2. Granular sludge EPS analysis and Fourier-infrared spectroscopy analysis

In general, EPS are commonly produced by microbes in response to unfavourable environments [38]. They typically are used as layers of protection for bacteria cells, which also enhances sludge particle aggregation and plays an important role in sludge granulation. Variation in the content and composition of LB-EPS and TB-EPS from granular sludge is shown in electronic supplementary material, figure S3a,b, respectively. Similar patterns of variation were observed for LB-EPS and TB-EPS. Specifically, LB-EPS and TB-EPS increased from 59.03 to 144.98 mg g$^{-1}$ VSS and from 116.05 to 214.09 mg g$^{-1}$ VSS from phase I to phase III, respectively, which indicated that salinity triggered EPS production and, in turn, enhanced sludge aggregation. Moreover, previous studies have shown that EPS play an important role in osmotic adaptation [39]. Bacterial cells that produce EPS can bind cations, experience reduced water loss in the cytoplasm and maintain a stable intracellular osmotic environment [40]. The positive EPS behaviours enable the granular sludge (AnAOB, DNB and SAD granules) to be embedded by EPS to counteract the osmotic pressure. PN/PS decreased from 15.58 to 10.05 and 22.59 to 13.32 for LB-EPS and TB-EPS, respectively. This result indicated that PS played an increasingly important role in osmoregulation.

Not surprisingly, LB-EPS and TB-EPS decreased to 95.04 and 170.26 mg g$^{-1}$ VSS, respectively, under high salinity (phase IV). This result stemmed from the weakening of the self-protection capability resulting from the death of bacteria cells not capable of tolerating high salinity. PN, the hydrophobic fraction of EPS, was previously reported to play an important role in sludge granulation [34]. The PN/PS ratio fell to its lowest value (4.83 for LB-EPS, 10.23 for TB-EPS) in this phase, which also explained the disintegration phenomenon of granular sludge. In phase V, the metabolic activities of surviving bacteria recovered, during which LB-EPS and TB-EPS increased to 111.34 and 220.94 mg g$^{-1}$ VSS, respectively. These favourable results promoted sludge re-aggregation and improved the particle size, which was consistent with the observed features of the sludge.

FT-IR analysis with the infrared spectra of LB-EPS and TB-EPS shown in electronic supplementary material, figure S3c,d, respectively, were used to evaluate variation in the EPS functional groups under different salinities. Similar profiles were achieved for both LB-EPS and TB-EPS, suggesting that salinity had a negligible impact on the consumption of EPS. Specifically, the broad band near 3406–3424 cm$^{-1}$ was assigned to the O–H stretching vibration in the hydroxyl groups. The slight adsorption at 2910–2955 cm$^{-1}$ was possibly formed by the C–H expansion vibration in alkane organic matter. The visible bands near 1622–1649 cm$^{-1}$ were produced by the stretching vibration of C=O and N–H (amide I vibration of the protein). The bands near 1411–1445 cm$^{-1}$ were probably attributed to the deformation vibration of C–OH, CH$_3$ and CH$_2$ in the protein structure. The C–N stretching vibration of protein amide III and the C–O–C stretching vibration of carbohydrate-like substances were observed near bands 1267–1273 cm$^{-1}$ and 1047–1094 cm$^{-1}$, respectively. These results indicated the EPS were mainly composed of protein-like and polysaccharide-like substances.

Overall, salinity had a more significant impact on the EPS content than the EPS composition. The contents of protein-like and polysaccharide-like substances in EPS varied at different salinities, which might be related to binding cations for the osmotic adaptation of functional bacteria.

**Table 2.** Diversity and richness indices of the microbial community.

| sample | phases | OTU | coverage | richness | | diversity | |
|---|---|---|---|---|---|---|---|
| | | | | Ace | Chao | Shannon | Simpson |
| S1 | I | 372 | 0.9987 | 460.5605 | 477.0000 | 3.4869 | 0.0594 |
| S2 | II | 486 | 0.9981 | 583.5079 | 557.3462 | 3.9082 | 0.0417 |
| S3 | III | 540 | 0.9983 | 610.1626 | 612.7500 | 3.9057 | 0.0496 |
| S4 | IV | 489 | 0.9984 | 540.0724 | 530.0000 | 3.5554 | 0.0905 |
| S5 | V | 429 | 0.9981 | 542.5133 | 534.0545 | 3.7493 | 0.0521 |

## 3.3. Analysis of microbial community structure and functional bacteria

### 3.3.1. Microbial diversity analysis

Illumina high-throughput sequencing and α-diversity analysis were used to characterize microbial community dynamics and shifts in microbial communities in the SAD process at different salinities. The diversity and richness results of microbial community structure are shown in table 2. A total of 299 885 effective sequences were obtained from five granular sludge samples, of which 61 203 sequences were used for taxonomic analysis. The coverage value of each sample was over 0.99, indicating that the results of the diversity analysis were robust. The OTUs, Ace and Chao values increased as the salinity concentration increased from S1 to S3, suggesting that salinity stimulated the growth of halophilic heterotrophic bacteria and thus increased the overall richness of the microbial community at low and medium salinity. The increased Shannon but decreased Simpson values suggested that microbial diversity increased as salinity increased. Specifically, OTUs, Ace, Chao and Shannon values decreased markedly, whereas the Simpson value increased sharply in S4, which indicated that the microbial richness and diversity significantly decreased at high salinity. This decrease in microbial richness and diversity probably stem from the elimination of bacterial species that could not endure high salinity stress. Finally, the diversity and richness indices recovered to some extent in S5 after the elimination of high salinity stress. This finding further suggested that the effect of high salinity on microbial community structure was reversible because of the recovered activity of surviving bacteria.

The similarity and discrepancy of microbial communities from the five granular sludge samples were further assessed through a Venn diagram analysis (electronic supplementary material, figure S4). A total of 227 OTUs were observed in all five granular sludge samples, and the number of unique OTUs was 2, 12, 52, 75 and 3 in S1, S2, S3, S4 and S5, respectively. The increase in unique OTUs as salinity increased revealed greater differences in microbial communities between sludge samples stemming from selection for salinity tolerance. The least common OTUs (257) and most unique OTUs (75) were detected between S1 and S4, further suggesting that high salinity had a significant impact on shifts in microbial communities in the SAD process.

### 3.3.2. Microbial community structure at the phylum and class levels

The microbial structure at the phylum and class levels (relative abundance (RA) > 0.01%) was further explored to gain insight into the population dynamics of the SAD process at different salinities (figure 3). The most common phyla in the five granular sludge samples were *Proteobacteria* (22.71–38.95%), *Acidobacteria* (11.33–17.08%), *Bacteroidetes* (7.67–21.22%), *Chloroflexi* (1.40–19.33%), *Planctomycetes* (3.21–16.41%), *Euryarchaeota* (1.10–11.12%), *Verrucomicrobia* (0.04–26.17%), *Armatimonadetes* (0.17–7.44%), *Patescibacteria* (0.29–2.00%) and *Gemmatimonadetes* (0.15–1.22%). *Proteobacteria* had the highest relative abundance among these 10 identified phyla throughout the entire operation of the reactor. Previous studies have confirmed that most of the heterotrophic bacteria capable of degrading organic matter belonged to *Proteobacteria* [11]. In addition, a few DNB were also classified within *Proteobacteria*. The high abundance of *Proteobacteria* might be explained by the following: (i) the growth of heterotrophic bacteria in the SAD process and (ii) the growth of halophilic heterotrophic bacteria because of salinity. *Bacteroidetes*, the RA of which increased as salinity increased, was associated with halophilic DNB [22].

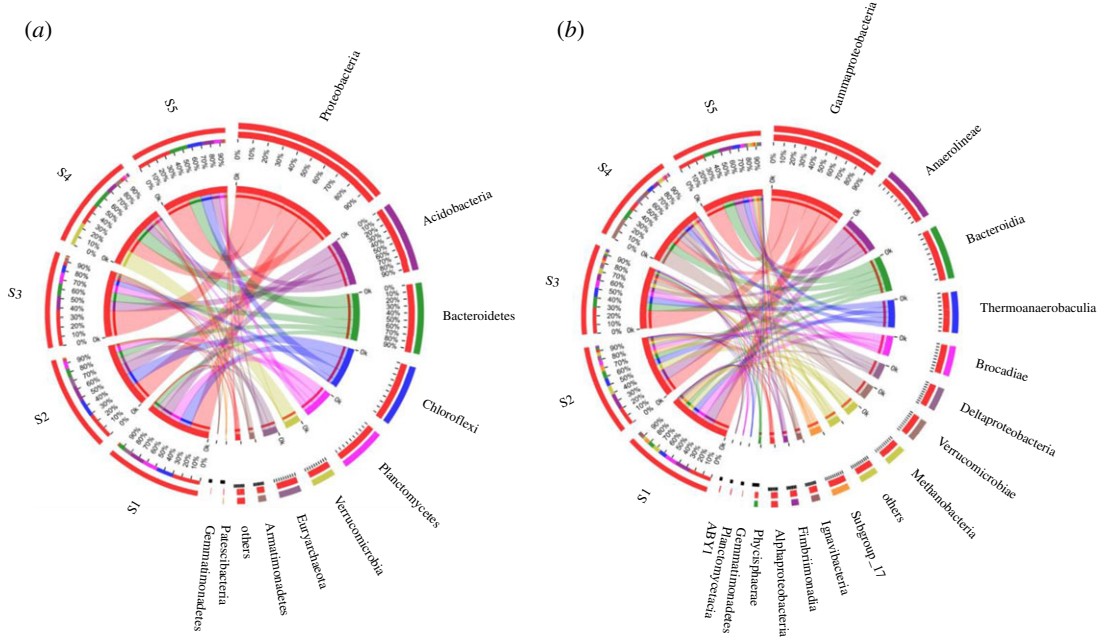

**Figure 3.** Per cent of community abundance at the (*a*) phylum and (*b*) class levels.

This favourable result promoted the denitrification pathway of the SAD process under saline conditions. *Acidobacteria* was maintained at a stable proportion and was closely associated with the biodegradation of low-weight molecular acids that were mainly derived from influent substrate [41]. In addition, the presence of *Euryarchaeota*, which were capable of producing methane under anaerobic conditions, also contributed to the complicated community structure of the microbial co-cultures.

*Planctomycetes*, which was previously reported to consist of all known AnAOB genera, provided a solid foundation for the anammox pathway under saline conditions. The pattern of the response of *Planctomycetes* to different salinities was complex. The RA of *Planctomycetes* first decreased and then increased from phase I to III, which probably reflected the adaptation of AnAOB under low and medium salinity. As expected, the lowest RA of *Planctomycetes* (3.21%) was observed at high salinity because of the elimination of AnAOB cells not tolerant of high salinity conditions. The increased proportion of *Planctomycetes* (7.43%) under non-saline conditions further reflected the survival of AnAOB cells after high salinity shock. *Chloroflexi* was found to coexist with *Planctomycetes*. This key phylum that was confirmed to form the skeleton of granular sludge played an important role in microbial aggregation. The low proportion of *Chloroflexi* (1.40%) could explain the disintegration of granular sludge after high salinity shock.

A total of 16 classes (RA > 0.01%) were identified in the five samples. The *γ-proteobacteria* (11.16–31.45%), *δ-proteobacteria* (2.78–9.92%) and *α-proteobacteria* (1.41–3.14%) classes were identified within the *Proteobacteria*. The *γ-proteobacteria* showed the highest RA in the microbial community at the class level during the operation of the reactor except during the high salinity period. Diverse DNB have been verified to belong to the *γ-proteobacteria* [42]. The RA of the *δ-proteobacteria* increased as salinity increased, which implied that salinity promoted the growth of halophilic heterotrophic bacteria. A similar increasing trend was observed for *Ignavibacteria*, which contributed to the increased RA of *Bacteroidetes* as salinity increased. The RA of *Thermoanaerobaculia* stabilized at approximately 9.35%, which explained the stable proportion of *Acidobacteria*. The proportion of *Methanobacteria*, which belonged to *Euryarchaeota*, decreased from 11.04% at low salinity to 1.05% at high salinity, indicating that salinity significantly inhibited Archaea. In addition, *Brocadiae* showed a similar pattern to *Planctomycetes*, suggesting that the AnAOB were included within this group. *Anaerolineae*, which was identified as the subgroup of *Chloroflexi*, had the lowest RA (0.95%) at high salinity, which further suggested that it played an important role in maintaining the structural integrity of the granular sludge.

### 3.3.3. Analysis of the microbial community at the genus level

To evaluate shifts in functional bacteria, eight identified genera (*Subgroup 10*, *Candidatus Jettenia*, *Methanobacterium*, *OLB12*, *OLB13*, *Denitratisoma*, *Ignavibacterium*, *Thauera* and *Ellin6067*) were screened

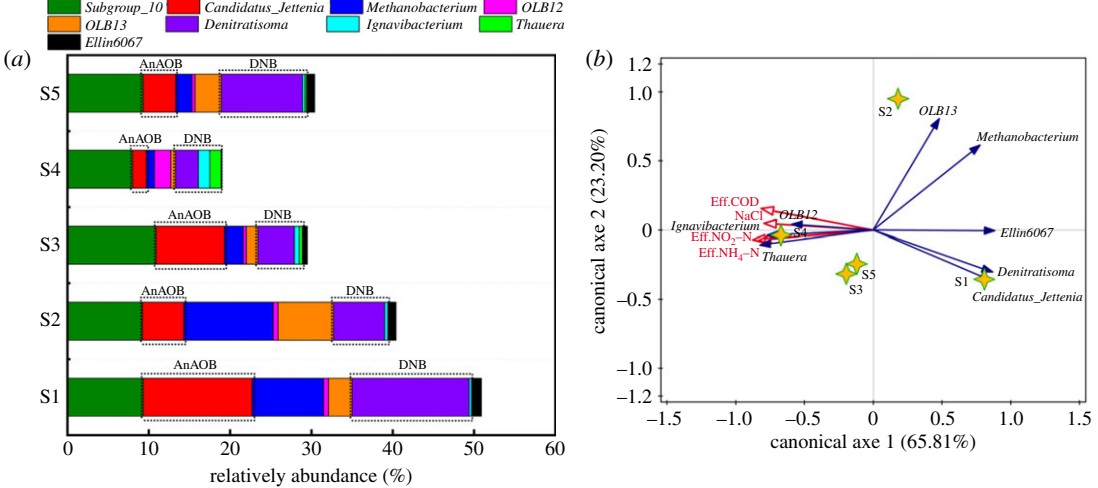

**Figure 4.** (*a*) Per cent of community abundance at the genus level and (*b*) correlation of environmental factors (Eff.COD, NaCl, Eff.NO$_2$–N and Eff.NH$_4$–N) and microbial communities based on a redundancy analysis (RDA).

out in the microbial phyla of the five samples (figure 4*a*). Salinity had a significant impact on the selection of the dominant functional genera for the anammox, denitrification and SAD processes (the details are shown in electronic supplementary material, table S4). *Candidatus Jettenia*, a functional genus of AnAOB, was responsible for the anammox pathway during the entire operation of the reactor. Nevertheless, variation in the RA of *Candidatus Jettenia* was relatively complex. The RA of *Candidatus Jettenia* decreased from 13.40 to 5.13% at low salinity primarily because of the inhibited activity, which induced even lower growth rates. Generally, heterotrophic bacteria showed superior salinity adaptation to autotrophic bacteria and could even be stimulated by the products of AnAOB under the anammox process [18]. The RA of *Candidatus Jettenia* increased to 8.42% under moderate salinity. This result was attributed to the fact that (i) the salinity adaptability of *Candidatus Jettenia* was enhanced after low salinity acclimation and that the (ii) double layers (outer EPS and inner DNB layers) protected *Candidatus Jettenia* from the toxic effects of salinity. A previous study found that EPS could behave as a selective membrane for regulating anion and cation concentrations and toxicity towards microorganisms [43], which thus combined with the DNB layer to maintain high activity of *Candidatus Jettenia*. Nevertheless, the RA of *Candidatus Jettenia* decreased to 1.68% under high salinity shock. A portion of *Candidatus Jettenia* was assumed to have been killed from the serious dehydration associated with high osmotic pressure. The disintegrated granular sludge with the decreased EPS revealed that *Candidatus Jettenia* was susceptible to direct exposure to Na$^+$ or Cl$^-$ and thus more vulnerable to salinity toxicity. The RA of *Candidatus Jettenia* later recovered to 4.01%, which was attributed to the ability of surviving *Candidatus Jettenia* to recover. These results could also explain the varying trends in both *Planctomycetes* and *Brocadiae*.

DNB were more diverse than AnAOB during the operation of the reactor. *Denitratisoma*, *Ignavibacterium* and *Thauera* were identified as the functional genera of DNB under different salinities. The pattern of variation in *Denitratisoma* was opposite to that of *Ignavibacterium* and *Thauera*. The highest relative abundance of *Denitratisoma* (14.37%) was observed in S1 when *Ignavibacterium* and *Thauera* were at their lowest RAs (0.30% and 0.01%, respectively). The RA of *Denitratisoma* decreased as salinity increased, and its RA was the lowest (2.80%) under high salinity. This result indicated that *Denitratisoma* was more vulnerable to salinity than *Candidatus Jettenia*, which possibly stemmed from its outer distribution around the granular sludge. Conversely, the proportion of *Ignavibacterium* and *Thauera* increased as salinity increased; final RAs of 1.45% and 1.44% were obtained under high salinity for *Ignavibacterium* and *Thauera*, respectively. This finding suggested that the halophilic DNB-*Ignavibacterium* and *Thauera* played increasingly important roles in denitrification under saline conditions, despite their proportions being lower than *Denitratisoma*. *Thauera* was previously reported to be a key microbial phylum for partial-denitrification treatment of saline wastewater [23]. This special physiological characterization of *Thauera* could explain the increased nitrite accumulation rate in §3.3, which helped enhance the coupling denitrification and anammox processes. The RA of *Denitratisoma* recovered to 9.99%, whereas the proportion of *Ignavibacterium* and *Thauera* decreased to 0.29% and 0.21%, respectively, when salinity stress was eliminated. These results further suggested

that the increased diversity of DNB supported the denitrification pathway and promoted the adaptation of the SAD process under salinity stress.

The correlations of environmental parameters with the microflora were further assessed using an RDA analysis (figure 4b). The red arrows represent the environmental parameters, and the blue arrows represent the identified genera. In addition, the length of the arrow implied the degree of influence of environmental parameters on the microflora. An angle between the two arrows below 90° indicates a positive correlation, whereas an angle above 90° indicates a negative correlation [44]. Salinity had a significant impact on the microbial community. All of the genera were negatively correlated with the environmental parameters with the exception of *Ignavibacterium* and *Thauera*, further confirming the halophilic physiology of these two bacteria. The negative correlation between NaCl with *Candidatus Jettenia* and *Denitratisoma* reflected the adverse effects of salinity on the SAD process. Eff.NH$_4$–N, Eff.NO$_2$–N and Eff.COD were negatively correlated with *Candidatus Jettenia* and *Denitratisoma*, which revealed the importance of the synergy of these two functional genera for the simultaneous removal of nitrogen and organic matter. Additionally, *Thauera* was positively correlated with Eff.NO$_2$–N, which indicated that it partially contributed to the denitrification behaviour. Thus, the increased salinity shock might alter the microbial community and induce shifts in the functional bacteria, resulting in a deterioration of the performance of the SAD process.

# 4. Conclusion

The SAD process showed moderate salinity resistance (less than or equal to 50 mmol l$^{-1}$) at a C/N ratio of 0.5, which resulted from the increased EPS content that promoted larger-size granule sludge. The stable removals of TN (84.58%) and COD (73.18%) could be finally achieved after low and moderate salinity acclimation. High salinity exposure disintegrated granular sludge and the subsequent collapse in the performance of the SAD process. Illumina-Miseq sequencing revealed that the diversity of DNB genera increased as salinity increased (including *Denitrisoma*, *Thauera* and *Ignavibacterium*), whereas an AnAOB genus (*Candidatus Jettenia*) remained constant during the entire operation of the reactor.

Data accessibility. Data available from the Dryad Digital Repository: https://doi.org/10.5061/dryad.kh189325h [45].

Authors' contributions. Z.W.: conceptualization, writing—review and editing, and supervision; Z.W., S.L., Y.J., P.G. and H.Z.: methodology; Z.W., S.L., P.G., Y.J., H.Z., X.W. and J.M.: formal analysis and investigation; Z.W. and P.G.: writing—original draft preparation. All authors gave final approval for publication.

Competing interests. The authors declare no competing interests.

Funding. This work was supported by Hebei Provincial innovation funding project for graduate students (CXZZSS2019071); Hebei Provincial Natural Science Fund Project (E2016402017); and Project of Young Top Talents Program in Universities and Colleges of Hebei Province (BJ2019029).

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
