## [Peer Review File · Royal Society Open Science]

Review History

RSOS-202099.R0 (Original submission)

Review form: Reviewer 1

Is the manuscript scientifically sound in its present form?

No

Are the interpretations and conclusions justified by the results?

Yes

Is the language acceptable?

Yes

Do you have any ethical concerns with this paper?

No

Have you any concerns about statistical analyses in this paper?

No

Recommendation?

Major revision is needed (please make suggestions in comments)

Comments to the Author(s)

The performance of biological nitrogen removal is susceptible to the variation of environmental factors. Salinity is a critical parameter in various water bodies. Most studies have indicated that the activity of microorganisms has a significant correlation with salinity. Except for halophilic bacteria, most bacteria are affected by different salinity for nitrogen-rich saline wastewater treatment. In this study, the authors revealed that high salinity reduced the removal efficiency of nitrogen and COD in a simultaneous anammox and denitrification(SAD) process. Nevertheless, the reactor could still maintain stable nitrogen removal efficiency under low and moderate salinity. This manuscript could provide a correct guideline for organic nitrogen-rich saline wastewater treatment. However, here are some questions that need to be addressed before consideration of publication.

1. The expression of “anaerobic anammox bacteria” was not standardized, anammox means anaerobic ammonia oxidation.
2. Line 19 of page 3, “substrate” should be plural.
3. It is necessary to cite references to prove the sentence of “In addition, nitrate, which accounts for approximately 10% of total nitrogen (TN), is produced after the anammox bioreaction.” In this text.
4. Line 10 of page 6, the unit of salinity should be consistent with the context.
5. Explain the meaning of “Eanammox,” and “Edenitrification”.
6. In fig.1(a), the salinity curve of the fourth stage is not prominent and needs further modification.
7. How to distinguish DNB granules and SAD granules with the same surface morphology.
8. Why choose sodium acetate as a carbon source.
9. “ammonia removal efficiency” should be changed to “NH₄⁺-N removal efficiency”, be consistent with the format of “NO₂⁻-N removal efficiency”.
10. The format of references needs further adjustment according to the journal guidelines.

Review form: Reviewer 2

Is the manuscript scientifically sound in its present form?

Yes

Are the interpretations and conclusions justified by the results?

Yes

Is the language acceptable?

Yes

Do you have any ethical concerns with this paper?

No

Have you any concerns about statistical analyses in this paper?

No

Recommendation?

Accept with minor revision (please list in comments)

Comments to the Author(s)

In this study, effects of salinity on the performance, sludge morphology, and microbial communities were studied in a simultaneous anammox and denitrification (SAD) process. The limitation of the process to the salinity upto 50 mmol/L was clearly showed and the recovery of the performance of the reactor from 100mmol to 0 mmol/L was also demonstrated. The effects of the salinity on the reactor performance were well analyzed by EPS and microbial analysis.

What is the typical salinity of saline wastewater such as pickle industrial wastewater? The tolerance of the SAD process to the salinity of 50mmol/L is enough for saline wastewater?

P8L8-14. How did you suppose the treatment system using the SAD for saline wastewater and design the synthetic wastewater containing ammonium and such high nitrite? How was dissolved oxygen of the synthetic wastewater? Partial nitrification using activated sludge is assumed as pretreatment of the SAD process?

P8L9-11. Define the C/N ratio.

P8L14-16. Describe more the history of the seed sludge, such as component of wastewater, temperature, biogas production. Three kinds of granules (AnAOB, DNB, and SAD) were intentionally mixed?

P11L5-6. How many grams of sludge collected? It contained 3 kinds of granules, AnAOB, DNB, and SAD?

P15L2-6. Particle sizes of 3 granules (AnAOB, DNB, and SAD) were different each other?

P16L4-6. The broken sludge was discharged from the reactor? Did you measure SS concentration?

P16L22 Fig,5b?

Fig.2. How was the size of the Petri dish?

Table S2. Definition of Eannamox was unclear.

Decision letter (RSOS-202099.R0)

Dear Dr Wang:

Title: Effects of salinity on the simultaneous anammox and denitrification process: performance, sludge morphology, and shifts in microbial communities

Manuscript ID: RSOS-202099

The editor assigned to your manuscript has now received comments from reviewers. We would like you to revise your paper in accordance with the referee and Subject Editor suggestions which can be found below (not including confidential reports to the Editor). Please note this decision does not guarantee eventual acceptance.

Please submit your revised paper before 05-Feb-2021. Please note that the revision deadline will expire at 00.00am on this date. If we do not hear from you within this time then it will be assumed that the paper has been withdrawn. In exceptional circumstances, extensions may be possible if agreed with the Editorial Office in advance. We do not allow multiple rounds of revision so we urge you to make every effort to fully address all of the comments at this stage. If deemed necessary by the Editors, your manuscript will be sent back to one or more of the original reviewers for assessment. If the original reviewers are not available we may invite new reviewers.

On behalf of the Subject Editor Professor Anthony Stace and the Associate Editor Dr Nadia Martinez Villegas.

RSC Associate Editor:

Comments to the Author:

The research presented in this draft is original and of interest to RSOS audience, however some clarification regarding different terms used in the manuscript as well as more detail in the materials and methods are needed. Please read carefully each of the comments from the reviewers and address each of them.

RSC Subject Editor:

Comments to the Author:

(There are no comments.)

Reviewers' Comments to Author:

Reviewer: 1

Comments to the Author(s)

The performance of biological nitrogen removal is susceptible to the variation of environmental factors. Salinity is a critical parameter in various water bodies. Most studies have indicated that the activity of microorganisms has a significant correlation with salinity. Except for halophilic bacteria, most bacteria are affected by different salinity for nitrogen-rich saline wastewater treatment. In this study, the authors revealed that high salinity reduced the removal efficiency of nitrogen and COD in a simultaneous anammox and denitrification(SAD) process. Nevertheless, the reactor could still maintain stable nitrogen removal efficiency under low and moderate salinity. This manuscript could provide a correct guideline for organic nitrogen-rich saline wastewater treatment. However, here are some questions that need to be addressed before consideration of publication.

1. The expression of "anaerobic anammox bacteria" was not standardized, anammox means anaerobic ammonia oxidation.
2. Line 19 of page 3, "substrate" should be plural.
3. It is necessary to cite references to prove the sentence of "In addition, nitrate, which accounts for approximately 10% of total nitrogen (TN), is produced after the anammox bioreaction." In this text.
4. Line 10 of page 6, the unit of salinity should be consistent with the context.
5. Explain the meaning of "Eanammox," and "Edenitrification".
6. In fig.1(a), the salinity curve of the fourth stage is not prominent and needs further modification.
7. How to distinguish DNB granules and SAD granules with the same surface morphology.
8. Why choose sodium acetate as a carbon source.
9. "ammonia removal efficiency" should be changed to "NH₄⁺-N removal efficiency", be consistent with the format of "NO₂⁻-N removal efficiency".
10. The format of references needs further adjustment according to the journal guidelines.

Reviewer: 2

Comments to the Author(s)

In this study, effects of salinity on the performance, sludge morphology, and microbial communities were studied in a simultaneous anammox and denitrification (SAD) process. The limitation of the process to the salinity upto 50 mmol/L was clearly showed and the recovery of the performance of the reactor from 100mmol to 0 mmol/L was also demonstrated. The effects of the salinity on the reactor performance were well analyzed by EPS and microbial analysis.

What is the typical salinity of saline wastewater such as pickle industrial wastewater? The tolerance of the SAD process to the salinity of 50mmol/L is enough for saline wastewater?

P8L8-14. How did you suppose the treatment system using the SAD for saline wastewater and design the synthetic wastewater containing ammonium and such high nitrite? How was dissolved oxygen of the synthetic wastewater? Partial nitrification using activated sludge is assumed as pretreatment of the SAD process?

P8L9-11. Define the C/N ratio.

P8L14-16. Describe more the history of the seed sludge, such as component of wastewater, temperature, biogas production. Three kinds of granules (AnAOB, DNB, and SAD) were intentionally mixed?

P11L5-6. How many grams of sludge collected? It contained 3 kinds of granules, AnAOB, DNB, and SAD?

P15L2-6. Particle sizes of 3 granules (AnAOB, DNB, and SAD) were different each other?

P16L4-6. The broken sludge was discharged from the reactor? Did you measure SS concentration?

P16L22 Fig.5b?

Fig.2. How was the size of the Petri dish?

Table S2. Definition of Eannamox was unclear.

Author's Response to Decision Letter for (RSOS-202099.R0)

See Appendix A.

RSOS-202099.R1 (Revision)

Review form: Reviewer 1

Is the manuscript scientifically sound in its present form?

Yes

Are the interpretations and conclusions justified by the results?

Yes

Is the language acceptable?

Yes

Do you have any ethical concerns with this paper?

No

Have you any concerns about statistical analyses in this paper?

No

Recommendation?

Accept as is

Comments to the Author(s)

All comments have been well responded.

Review form: Reviewer 2

Is the manuscript scientifically sound in its present form?

Yes

Are the interpretations and conclusions justified by the results?

Yes

Is the language acceptable?

Yes

Do you have any ethical concerns with this paper?

No

Have you any concerns about statistical analyses in this paper?

Yes

Recommendation?

Accept with minor revision (please list in comments)

Comments to the Author(s)

Authors answered and commented well to my reviews.

For Comment 1, please reflect your answers to the manuscript, eg. pp4-5.

For Comment 7, please reflect your answer to the manuscript, eg. pp17-18.

Decision letter (RSOS-202099.R1)

Dear Dr Wang:

Title: Effects of salinity on the simultaneous anammox and denitrification process: performance, sludge morphology, and shifts in microbial communities

Manuscript ID: RSOS-202099.R1

Thank you for submitting the above manuscript to Royal Society Open Science. On behalf of the Editors and the Royal Society of Chemistry, I am pleased to inform you that your manuscript will be accepted for publication in Royal Society Open Science subject to minor revision in accordance with the referee suggestions. Please find the reviewers' comments at the end of this email.

The reviewers and handling editors have recommended publication, but also suggest some minor revisions to your manuscript. Therefore, I invite you to respond to the comments and revise your manuscript.

Because the schedule for publication is very tight, it is a condition of publication that you submit the revised version of your manuscript before 31-Mar-2021. Please note that the revision deadline will expire at 00.00am on this date. If you do not think you will be able to meet this date please let me know immediately.

Kind regards,
Dr Laura Smith
Publishing Editor, Journals

Royal Society of Chemistry
Thomas Graham House

Science Park, Milton Road
Cambridge, CB4 0WF
Royal Society Open Science - Chemistry Editorial Office

On behalf of the Subject Editor Professor Anthony Stace and the Associate Editor Dr Nadia Martinez Villegas.

RSC Associate Editor:
Comments to the Author:
The authors have satisfactorily addressed all reviewer’s comments. The manuscript can now be accepted with minor revisions.

RSC Associate Editor:
Comments to the Author:
(There are no comments.)

Reviewer comments to Author:
Reviewer: 1

Comments to the Author(s)
All comments have been well responded.

Reviewer: 2

Comments to the Author(s)
Authors answered and commented well to my reviews.
For Comment 1, please reflect your answers to the manuscript, eg. pp4-5.
For Comment 7, please reflect your answer to the manuscript, eg. pp17-18.

Author's Response to Decision Letter for (RSOS-202099.R1)

See Appendices B & C.

Decision letter (RSOS-202099.R2)

Dear Dr Wang:

Title: Effects of salinity on the simultaneous anammox and denitrification process: performance, sludge morphology, and shifts in microbial communities
Manuscript ID: RSOS-202099.R2

It is a pleasure to accept your manuscript in its current form for publication in Royal Society Open Science. The chemistry content of Royal Society Open Science is published in collaboration with the Royal Society of Chemistry.

On behalf of the Subject Editor Professor Anthony Stace and the Associate Editor Dr Nadia Martinez Villegas.

RSC Associate Editor
Comments to the Author:
The authors have addressed all necessary comments and remarks from the reviewers. The manuscript can now be accepted as is.

Reviewer(s)' Comments to Author:

Appendix A

Dear Laura Smith:

Thank you so much for giving us the opportunity to revise and resubmit our manuscript (RSOS-202099), entitled 'Effects of salinity on the simultaneous anammox and denitrification process: performance, sludge morphology, and shifts in microbial communities'. We sincerely thank you and the reviewer for your valuable feedback that we have used to improve the quality of our manuscript. The reviewer comments are laid out below in italicized font and specific concerns have been numbered. Our response is given in normal font and changes/additions to the manuscript are given in yellow text.

We hope that the revised version of the manuscript could be considered for publication in your journal. I look forward to hearing from you soon.

With best wishes,

Yours sincerely,

Zhaozhao Wang,

Jan 25, 2021

To reviewer:

We sincerely thank you for your professional review work on our manuscript. Your thoughtful comments and constructive suggestions have contributed a lot to improve the quality of our manuscript. As you are concerned, there are several problems that need to be addressed. According to your nice suggestions, we have made extensive corrections to our previous draft. After this revision, we have written a point-by-point response letter to you as you can see above. And the detailed corrections are listed below.

Reviewer 1

Comment 1: *The expression of “anaerobic anammox bacteria” was not standardized, anammox means anaerobic ammonia oxidation.*

Response: We sincerely thank you for your professional review work. We have checked and corrected the expression of “anaerobic anammox bacteria” in the article, which has been corrected to “anaerobic ammonia-oxidizing bacteria” in the article. The corrected content is as follows:

Line 15 of page 3.

“Anammox is an autotrophic nitrogen removal process occurring under anaerobic conditions that is dependent on a class of **anaerobic ammonia-oxidizing bacteria** (AnAOB) belonging to the phylum *Planctomycetes*, which is known to include five functional genera: *Ca. Brocadia*, *Ca. Kuenenia*, *Ca. Scalindua*, *Ca. Anammoxoglobus*, and *Ca. Jettenia* ^[5,6].”

Many thanks again!

Comment 2: *Line 19 of page 3, “substrate” should be plural.*

Response: Thank you for your careful check. We have checked and corrected the grammar of “substrate” in the article, which has been corrected to “substrates” in the article. The corrected content is as follows:

Line 19 of page 3.

“Nevertheless, AnAOB have a slow growth rate and can be affected by both **substrates** and environmental factors, among which organic matter is extremely important. “

Many thanks again!

Comment 3: *It is necessary to cite references to prove the sentence of “In addition, nitrate, which accounts for approximately 10% of total nitrogen (TN), is produced after the anammox bioreaction.” In this text.*

Response: We sincerely thank you for your professional review work. We have added a supporting literature at the corresponding position in the article, the content is as follows:

Line 2-3 of page 4.

“In addition, nitrate, which accounts for approximately 10% of total nitrogen (TN), is produced after the anammox bioreaction **[9]**.”

[9] Chen HH, Liu ST, Yang FL, Xue Y, Wang T. 2009 The development of simultaneous partial nitrification, ANAMMOX and denitrification (SNAD) process in a single reactor for nitrogen removal. *Bioresource Technology*. 100, 1548–1554. (doi:10.1016/j.biortech.2008.09.003)”

Many thanks again!

Comment 4: *Line 10 of page 6, the unit of salinity should be consistent with the context.*

Response: We sincerely thank you for your professional review work. We have unified the units at the corresponding positions in the text, and the corrections are as follows:

Line 10 of page 6.

“Zhai et al. [21] found that the denitrification efficacy decreased as a result of the inhibition of microorganismal activities by high salinity (**598.9 mmol/L**);”

Many thanks again!

Comment 5: Explain the meaning of “Eanammox,” and “Edenitrification”.

Response: We sincerely thank you for your professional review work. We have supplemented the meaning of “ $E_{anammox}$ ” and “ $E_{denitrification}$ ” in the corresponding positions in the text, and the supplementary content is as follows:

Line 11-13 of page 9.

“The $\text{NH}_4^+\text{-N}$ removal efficiency (ARE), nitrogen removal efficiency (NRE), $\text{NO}_2^-\text{-N}$ removal efficiency, COD removal efficiency (CRE), nitrogen loading rate (NLR), nitrogen removal rate (NRR), COD removal rate (CRR), $\Delta\text{NO}_2^-\text{-N}/\Delta\text{NH}_4^+\text{-N}$ ratio, $\Delta\text{NO}_3^-\text{-N}/\Delta\text{NH}_4^+\text{-N}$ ratio, the contribution of nitrogen removal via the anammox process ($E_{anammox}$), and the contribution of nitrogen removal via the denitrification process ($E_{denitrification}$) were calculated according to the formulas in Table S2.”

Many thanks again!

Comment 6: In fig.1(a), the salinity curve of the fourth stage is not prominent and needs further modification.

Response: We sincerely thank you for your professional review work. We have modified the salinity curve of the fourth stage in Fig.1(a) to make it more obvious. Many thanks again!

Fig.1. Performance of the SAD process during different operational phases. (a) ammonia, nitrite, and nitrate removals; (b) TN removal; (c) COD removal; and (d) changes in $\Delta\text{NO}_2^-\text{-N}/\Delta\text{NH}_4^+\text{-N}$ and $\Delta\text{NO}_3^-\text{-N}/\Delta\text{NH}_4^+\text{-N}$.

Comment 7: *How to distinguish DNB granules and SAD granules with the same surface morphology.*

Response: We sincerely thank you for your professional review work. Regarding on the distinctions between DNB granules and SAD granules: Firstly, the average particle size of the SAD granule is relatively larger than that of the DNB granule; secondly, the surface and interior colors of the DNB granule are gray white, whereas it is outer-white and inner-dark red for the SAD granule . We hope our explanation can satisfy you.

Many thanks again!

Comment 8: *Why choose sodium acetate as a carbon source.*

Response: We sincerely thank you for your professional review work. While sodium acetate is one of the most commonly used organic carbon sources in the sewage treatment field; moreover, choosing this carbon source (e.g. unselective of other carbon source, such as glucose, et al.) can avoid the confusions with the carbohydrate or protein of EPS and SMP detections in this study. We hope our explanation can satisfy you.

Many thanks again!

Comment 9: *“ammonia removal efficiency” should be changed to “NH₄⁺-N removal efficiency”, be consistent with the format of “NO₂⁻-N removal efficiency”.*

Response: We sincerely thank you for your professional review work. We have changed the format of “ammonia removal efficiency” in the text, and the changes are as follows:

Line 8 of page 9.

“The **NH₄⁺-N removal efficiency** (ARE), nitrogen removal efficiency (NRE), NO₂⁻-N removal efficiency, COD removal efficiency (CRE), nitrogen loading rate (NLR), nitrogen removal rate (NRR), COD removal rate (CRR), $\Delta\text{NO}_2^- \text{-N} / \Delta\text{NH}_4^+ \text{-N}$ ratio, $\Delta\text{NO}_3^- \text{-N} / \Delta\text{NH}_4^+ \text{-N}$ ratio, **the contribution of nitrogen removal via the anammox process (E_{anammox})**, and **the contribution of nitrogen removal via the denitrification process ($E_{\text{denitrification}}$)** were calculated according to the formulas in Table S2.”

Many thanks again!

Comment 10: *The format of references needs further adjustment according to the journal guidelines.*

Response: We sincerely thank you for your professional review work. We have carefully checked and corrected the formatting of all the references.

Many thanks again!

Reviewer 2

Comment 1: *What is the typical salinity of saline wastewater such as pickle industrial wastewater? The tolerance of the SAD process to the salinity of 50mmol/L is enough for saline wastewater?*

Response: We sincerely thank you for your professional review work. The salinity range of saline wastewater (such as pickle industrial wastewater) is generally 170-2500 mmol/L, while the dilution is generally used before the biological treatment process. In this study, it was found that the SAD process showed a relative sensitive tolerance of 50 mmol/L, indicating that the SAD process can be applied in the real saline wastewater treatment with an appropriate dilution of the feeding sewage if the original salinity is too high. We hope our explanation can satisfy you.

Many thanks again!

Comment 2: *P8L8-14. How did you suppose the treatment system using the SAD for saline wastewater and design the synthetic wastewater containing ammonium and such high nitrite? How was dissolved oxygen of the synthetic wastewater? Partial nitrification using activated sludge is assumed as pretreatment of the SAD process?*

Response: We sincerely thank you for your professional review work.

1) Compared with the traditional biological nitrogen process, the SAD process is a higher-efficient and more economical sewage treatment method by the coupling of anammox and denitrification processes. However, there are few reports on the treatment of salt-containing organic nitrogen-containing wastewater by the SAD process. Therefore, this study explores the influences of salinity on the SAD process to widen its application. As the requirement for the anammox reaction, ammonium and nitrite were added in the synthetic wastewater and their concentrations were the same as those for the seeding sludge reactor, which intended to treat a relative high nitrogen-containing sewage.

2) The synthetic wastewater was in a relative lower dissolved oxygen (DO) state (DO concentration was in the range of 0-1.0 mg/L). We have supplemented the DO concentration of synthetic wastewater at the corresponding part of the article, which was shown as:

Line 9 of page 8.

“Synthetic wastewater (DO: 0-1.0 mg/L) was used as the feeding substrate; the ammonia, nitrite, organic matter, and salinity originated from NH_4Cl , NaNO_2 , sodium acetate, and NaCl , respectively.”

3) For the SAD process, both of the ammonium and nitrite were needed in the feeding sewage. If the original nitrogen in the feed existed only as the ammonium, and the partial nitrification is needed as the pretreatment for the SAD process. Moreover, other methods could also be utilized to satisfy the feed compositions by an appropriate mixing of the wastewater that only has ammonium or nitrite as the sole nitrogen source. We hope our explanation can satisfy you.

Many thanks again!

Comment 3: *P8L9-11. Define the C/N ratio.*

Response: We sincerely thank you for your professional review work. We have made a supplementary explanation of the meaning of “C/N ratio” at the corresponding position in the text, and the supplementary content is as follows:

Line 7-8 of page 9.

“The C/N ratio was defined as the chemical oxygen demand (COD)/NO_x-N ratio in the influent.”

Many thanks again!

Comment 4: *Describe more the history of the seed sludge, such as component of wastewater, temperature, biogas production. Three kinds of granules (AnAOB, DNB, and SAD) were intentionally mixed?*

Response: We sincerely thank you for your professional review work. We have added the history description of the seed sludge in the article, and the added content is as follows:

Line 14-22 of page 8.

“The seed granular sludge was obtained from a pilot-scale SAD-UASB reactor. The SAD-UASB reactor was successfully shifted from an Anammox-UASB reactor as the following procedures: At 30°C, the anammox granular sludge were inoculated in the UASB reactor; after that, carbon source (sodium acetate) was introduced in the UASB system to cultivate the SAD granular sludge; finally, the SAD process was successfully started up by controlling the COD/NO_x-N ratio of 0.5 with the formations of three kinds of granules (anammox, DNB and SAD) in the bottom of the UASB reactor. The SAD-UASB was used for the treatment of high-nitrogen organic wastewater (NH₄⁺-N: 108.2 mg/L; NO₂⁻-N: 152.0 mg/L; NO₃⁻-N: 14.3 mg/L; COD: 84.1 mg/L) with an average N₂ production of 20 L/d and had been operated stably for half a year.”

Many thanks again!

Comment 5: *P11L5-6. How many grams of sludge collected? It contained 3 kinds of granules, AnAOB, DNB, and SAD?*

Response: We sincerely thank you for your professional review work. We are sorry for that we have not weighed the exact collected granules. For the molecular biological analysis, we took the mixed granules (10 ml) from the bottom of the UASB reactor, which were enough for the analysis amount. We have added this content at the corresponding position in the article, which is shown as follows:

Line 16 of page 11.

“Sludge samples (10 mL for each sample) were collected at the end of each phase, and a total of 5 samples were collected (S1, S2, S3, S4, and S5).”

Many thanks again!

Comment 6: *P15L2-6. Particle sizes of 3 granules (AnAOB, DNB, and SAD) were different each other?*

Response: We sincerely thank you for your professional review work. The dominated granular sludge in this study is SAD granule, and the scattered AnAOB and DNB granules are relatively less. One the whole, the particle size of SAD granule is relatively larger than that of AnAOB granule and DNB granule. We hope our explanation can satisfy you.

Many thanks again!

Comment 7: *P16L4-6. The broken sludge was discharged from the reactor? Did you measure SS concentration?*

Response: We sincerely thank you for your professional review work. In this study, due to the high salinity shocking, the granular sludge have been cracked into smaller broken particles, of which the average particle size were larger than that of typical sludge flocs. Owing to the effect of the gravity and the rising flow rate, the broken granular sludge could sink or be suspended in the the UASB reactor. The broken granular sludge still contained a large number of functional bacteria, so these broken sludge were not discharged artificially. Since the sludge concentration in the system was relatively stable, so no SS concentration measurement is performed. We hope our explanation can satisfy you.

Many thanks again!

Comment 8: *P16L22 Fig.5b?*

Response: We sincerely thank you for your professional review work. We have corrected the error of the figure number in the text, and the correction content is as follows:

Line 12 of page 17.

“Variation in the content and composition of LB-EPS and TB-EPS from granular sludge is shown in Fig. S3a and S3b, respectively.”

Many thanks again!

Comment 9: *Fig.2. How was the size of the Petri dish?*

Response: We sincerely thank you for your professional review work. The diameter of the petri dish in this study is 90 mm. We have made a supplementary explanation in the article, and the supplementary content is as follows:

Line 17-18 of page 9.

“The granular sludge was placed in a culture dish (90 mm) to observe the apparent sludge morphology.”

Many thanks again!

Comment 10: *Table S2. Definition of E_{anammox} was unclear.*

Response: We sincerely thank you for your professional review work. We have added the explanation of “E_{anammox}” at the corresponding position in the text, and the added content is as follows:

Line 12 of page 9.

“The NH₄⁺-N removal efficiency (ARE), nitrogen removal efficiency (NRE), NO₂⁻-N removal efficiency, COD removal efficiency (CRE), nitrogen loading rate (NLR), nitrogen removal rate (NRR), COD removal rate (CRR), ΔNO₂⁻-N/ΔNH₄⁺-N ratio, ΔNO₃⁻-N/ΔNH₄⁺-N ratio, the contribution of nitrogen removal via the anammox process (E_{anammox}), and the contribution of nitrogen removal via the denitrification process (E_{denitrification}) were calculated according to the formulas in Table S2.”

Many thanks again!

Appendix B

Dear editors:

Thank you very much for your acceptance of our paper (rsos-202099.R2), entitled 'Effects of salinity on the simultaneous anammox and denitrification process: performance, sludge morphology, and shifts in microbial communities'.

We have corrected the uploading files according to the journal's requirements. Now we have uploaded the supplementary figures and tables within only one file of "Electronic supplementary materials". We have completed the "Details & Comments" section. And we uploaded the raw data involved to Dryad Digital Repository (Dryad doi:10.5061/dryad.kh189325h. [Wang et al (2021)]). Besides, we have restored the category back to Chemistry.

If you have any questions at all, please do not hesitate to get in touch.

With best wishes,

Yours sincerely,

Zhaozhao Wang,

Mar, 2021

To reviewer

We sincerely thank you for your professional review work on our manuscript.

Your thoughtful comments and constructive suggestions have contributed a lot to improve the quality of our manuscript. According to your nice suggestions, we have made extensive corrections to our previous draft. After this revision, we have written a point-by-point response letter to you as you can see above. And the detailed corrections are listed below. Many thanks!

To reviewer 2

For Comment 1, please reflect your answers to the manuscript, eg. pp4-5.

Response: We sincerely thank you for your professional review work. We have already made supplements in the article, and the supplementary content is as follows:

Line 2-3 and 6-7 of page 5.

“Some types of wastewater contain salt, (e.g. pickle industrial wastewater [12], seafood industrial wastewater [13], of which the salinity concentration is normally in the range of 170-2500 mmol/L), which creates another challenge during the SAD process. High salt content places stress on the microbial flora, inhibits the activity of key enzymes, and eventually leads to cell disintegration and death [14]. In addition, appropriate acclimation strategies (e.g. dilution of the original saline wastewater, salinity-step domestication, etc.) can be used to permit the biological process to effectively treat saline wastewater under the salinity threshold.”

Many thanks again!

For Comment 7, please reflect your answer to the manuscript, eg. pp17-18.

Response: We sincerely thank you for your professional review work. We have already made supplements in the article, and the supplementary content is as follows:

Line 3-5 of page 17.

“Moreover, the proportion of granular sludge with particle sizes over 1.5 mm accounted for 83.74% of all particles, which revealed that the disintegration of the granular sludge resulting from the strong salinity shock was recoverable. It should be noted that no sludge was discharged out of the reactor throughout the experiment, which resulted in a relatively stable sludge concentration, and thus supported the recovery performance of the system.”

Many thanks again!

Appendix C

Dear Laura Smith:

Thank you very much for your acceptance of our paper (rsos-202099.R2), entitled ‘Effects of salinity on the simultaneous anammox and denitrification process: performance, sludge morphology, and shifts in microbial communities’.

We have modified and uploaded according to the upload requirements in the email. We have only uploaded the supplementary graphics and forms as "electronic supplementary materials" files. We have added the heading and legend of the supplementary file at the "Details & Comments" step of the online form. And, We also have restored the category back to Chemistry. At the same time, we uploaded the test data involved to Dryad Digital Repository (Dryad doi:10.5061/dryad.kh189325h. [Wang et al (2021)]).

Thank you again for your acceptance of our paper! If you have any questions at all, please do not hesitate to get in touch.

With best wishes,

Yours sincerely,

Zhaozhao Wang,
Mar, 2021

To reviewer

We sincerely thank you for your professional review work on our manuscript.

Your thoughtful comments and constructive suggestions have contributed a lot to improve the quality of our manuscript. According to your nice suggestions, we have made extensive corrections to our previous draft. After this revision, we have written a point-by-point response letter to you as you can see above. And the detailed corrections are listed below. Many thanks!

To reviewer 2

For Comment 1, please reflect your answers to the manuscript, eg. pp4-5.

Response: We sincerely thank you for your professional review work. We have already made supplements in the article, and the supplementary content is as follows:

Line 2-3 and 6-7 of page 5.

“Some types of wastewater contain salt, (e.g. pickle industrial wastewater [12], seafood industrial wastewater [13], of which the salinity concentration is normally in the range of 170-2500 mmol/L), which creates another challenge during the SAD process. High salt content places stress on the microbial flora, inhibits the activity of key enzymes, and eventually leads to cell disintegration and death [14]. In addition, appropriate acclimation strategies (e.g. dilution of the original saline wastewater, salinity-step domestication, etc.) can be used to permit the biological process to effectively treat saline wastewater under the salinity threshold.”

Many thanks again!

For Comment 7, please reflect your answer to the manuscript, eg. pp17-18.

Response: We sincerely thank you for your professional review work. We have already made supplements in the article, and the supplementary content is as follows:

Line 3-5 of page 17.

“Moreover, the proportion of granular sludge with particle sizes over 1.5 mm accounted for 83.74% of all particles, which revealed that the disintegration of the granular sludge resulting from the strong salinity shock was recoverable. It should be noted that no sludge was discharged out of the reactor throughout the experiment, which resulted in a relatively stable sludge concentration, and thus supported the recovery performance of the system.

Many thanks again!